# Glycochenodeoxycholate Affects Iron Homeostasis via Up-Regulating Hepcidin Expression

**DOI:** 10.3390/nu14153176

**Published:** 2022-08-02

**Authors:** Long-jiao Wang, Guo-ping Zhao, Xi-fan Wang, Xiao-xue Liu, Yi-xuan Li, Li-li Qiu, Xiao-yu Wang, Fa-zheng Ren

**Affiliations:** 1Key Laboratory of Functional Dairy, Co-Constructed by Ministry of Education and Beijing Municipality, College of Food Science & Nutritional Engineering, China Agricultural University, Beijing 100083, China; wanglongjiao@cau.edu.cn (L.-j.W.); wangxfan@126.com (X.-f.W.); b20183060508@cau.edu.cn (X.-x.L.); qiulily@cau.edu.cn (L.-l.Q.); 2School of Food and Health, Beijing Technology and Business University, Beijing 100048, China; zhaogp@btbu.edu.cn; 3Key Laboratory of Precision Nutrition and Food Quality, Department of Nutrition and Health, China Agricultural University, Beijing 100083, China; liyixuan@cau.edu.cn

**Keywords:** glycochenodeoxycholate, hepcidin, iron homeostasis, farnesoid X receptor, SMAD1/5/8

## Abstract

Increasing hepcidin expression is a vital factor in iron homeostasis imbalance among patients with chronic kidney disease (CKD). Recent studies have elucidated that abnormal serum steroid levels might cause the elevation of hepcidin. Glycochenodeoxycholate (GCDCA), a steroid, is significantly elevated in patients with CKD. However, the correlation between GCDCA and hepcidin has not been elucidated. Decreased serum iron levels and increased hepcidin levels were both detected in patients with CKD in this study. Additionally, the concentrations of GCDCA in nephropathy patients were found to be higher than those in healthy subjects. HepG2 cells were used to investigate the effect of GCDCA on hepcidin in vitro. The results showed that hepcidin expression increased by nearly two-fold against control under 200 μM GCDCA treatment. The phosphorylation of SMAD1/5/8 increased remarkably, while STAT3 and CREBH remained unchanged. GCDCA triggered the expression of farnesoid X receptor (FXR), followed with the transcription and expression of both BMP6 and ALK3 (upward regulators of SMAD1/5/8). Thus, GCDCA is a potential regulator for hepcidin, which possibly acts by triggering FXR and the BMP6/ALK3-SMAD signaling pathway. Furthermore, 40 C57/BL6 mice were treated with 100 mg/kg/d, 200 mg/kg/d, and 300 mg/kg/d GCDCA to investigate its effect on hepcidin in vivo. The serum level of hepcidin increased in mice treated with 200 mg/kg/d and 300 mg/kg/d GCDCA, while hemoglobin and serum iron levels decreased. Similarly, the FXR-mediated SMAD signaling pathway was also responsible for activating hepcidin in liver. Overall, it was concluded that GCDCA could induce the expression of hepcidin and reduce serum iron level, in which FXR activation-related SMAD signaling was the main target for GCDCA. Thus, abnormal GCDCA level indicates a potential risk of iron homeostasis imbalance.

## 1. Introduction

Hepcidin is a liver-expressed antimicrobial peptide (encoded by the Hamp gene) which is directly involved in iron metabolism. Furthermore, hepcidin serves as an important mediator in the pathogenesis of iron-related chronic diseases [1], and high levels of hepcidin (pro-hepcidin) are found in patients with anemia due to chronic disease [2]. Patients with chronic kidney disease (CKD) have abnormally high levels of serum hepcidin, which is correlated with the imbalance of iron homeostasis and patient survival [3,4]. It is generally recognized that inflammation and erythropoietin deficiency induce elevated expression of hepcidin. Signal transducer and activator of transcription 3 (STAT3) phosphorylation (mediated by inflammation), small mothers against decapentaplegic-1/5/8 (SMAD1/5/8) phosphorylation, and signaling regulated by nuclear transcription factors (such as cyclic AMP-responsive element-binding protein 3-like protein 3, also named CREBH) are the main regulatory pathways for hepcidin expression. However, treatment targeting these pathways could not fully counteract the up-regulated hepcidin, indicating that other, unknown factors exist for hepcidin regulation.

Bile acids are amphipathic steroid molecules synthesized in hepatocytes, whose serum levels are disturbed in patients with CKD. Recent studies have indicated a significant correlation between serum bile acids concentration and the concentration of hemoglobin [5,6]. Glycochenodeoxycholate (GCDCA) is the most abundant conjugated bile acid in humans which increases significantly in CKD. GCDCA is an important signaling molecule associated with many diseases and signaling pathways. It has been reported that GCDCA is related with hepcidin expression in rat model [7]. In addition, optimal bile acid supplementation (whole bile acids including GCDCA) increased hepcidin expression in fish [8]. Thus, we speculated that GCDCA might play an important role in hepcidin regulation.

However, there are no comprehensive studies on the relationship between GCDCA and hepcidin, as well as the underlying mechanism. The purpose of this study was to clarify the effect of GCDCA on hepcidin to provide fundamental basis for the regulation of iron homeostasis.

## 2. Materials and Methods

### 2.1. Determination of Population Experimental Indexes

A total of 223 patients with CKD were recruited from four hemodialysis centers in Beijing, China. The experiment was approved by the Ethics Committee of China Space Center Hospital (20151230-YW-01) and Ethics Committee of China Agricultural University (201503). All participants were diagnosed with CKD. The control group included 69 healthy individuals. Serum samples were isolated and stored at −80 °C. Concentrations of serum bile acids were detected by targeted liquid chromatography, according to the method reported by Wang et al. [5]. The iron related indexes were determined using biochemical analyzer. The indexes of blood routine were measured by blood routine analyzer. Since the elevation of hepcidin in CKD patients has been reported before [9,10,11], a count of 30 samples (including 10 healthy subjects and 20 CKD patients) were tested for serum hepcidin level in this study. Determination of serum hepcidin was performed using ELISA in 96-well plates coated with a polyclonal antibody against hepcidin (R&D Sys-tems, Minneapolis, MN, USA).

### 2.2. Chemicals

GCDCA (Purity 97%) was purchased from Shanghai Yuanye Bio-Technology Company (Shanghai, China). It was dissolved in cell medium and stored at 4 °C before used. LDN-193189 and GW4064 were purchased from Selleck Chemicals Company (Houston, TX, USA). Guggulsterone was purchased from MedChemExpress Company (Shanghai, China). LDN-193189, GW4064 and guggulsterone were dissolved in DMSO and stored at −20 °C before used.

### 2.3. Cell Culture

Human hepatocellular liver carcinoma cell line (HepG2) was cultured in minimum essential medium, supplemented with 10% fetal bovine serum, 1% glutamax, 1% non-essential amino acids (100×), and 1 mM sodium pyruvate solution. Above reagents were purchased from Gibco (Grand Island, NY, USA). In addition, 1% penicillin-streptomycin (Beyotime Biotechnology, Beijing, China) was also added into medium. Cells were cultured at 37 °C under 5% CO_2_ environment.

### 2.4. Cell Viability Determination

HepG2 cells were seeded in 96-well plates at 5000 cells per well. After cells attached to the surface and grew to about half of the well, different concentrations of GCDCA (0, 10 µM, 50 µM, 100 µM, 200 µM, 300 µM, 500 µM, 750 µM and 1000 µM) were used to treat cells for 24 h. Then, the cell viabilities in different treatment groups were determined using cell counting kit-8 assay (Beyotime Biotechnology, Beijing, China).

### 2.5. Quantitative Real-Time Polymerase Chain Reaction Analysis

Total RNA was obtained by TRIzol solution (Invitrogen, Carlsbad, CA, USA). Then it was reverse-transcribed by 5× All-In-One RT Master Mix (Applied Biological Materials, Inc., Richmond, BC, Canada). The contents of mRNA were determined by real-time quantitative polymerase chain reaction (RT-qPCR) on a 7900HT instrument (Applied Biosystems, Forster, CA, USA) using TB Green Premix Ex Taq (Takara Biomedical Technology Co., Ltd., Shiga, Japan) following the manufacturers’ protocol. The sequences of primers were shown in Appendix A.

### 2.6. Western Blotting Analysis

Lysates from HepG2 cells and mice liver and duodenum tissues were dissolved in the RIPA buffer containing protease inhibitor. The protein contents were quantified using BCA protein assay kit (Thermo Scientific, Waltham, MA, USA). Western blotting was used to evaluate protein expression by incubating the blots with different antibodies. Primary antibodies rabbit anti-pSMAD1/5/8 (1:2000, #13820), rabbit anti-SMAD1 (1:2000, #6944), rabbit anti-pSTAT3 Tyr705 (1:2000, #9145), rabbit anti-STAT3 (1:2000, #4904) and mouse anti-GAPDH (1:4000, #97166) were purchased from Cell Signaling Technology (Danvers, MA, USA). Primary antibodies rabbit anti-bone morphogenetic protein 2 (BMP2, 1:1000, YT5651), rabbit anti-BMP4 (1:1000, YT7841), rabbit anti-BMP6 (1:1000, YT0502), rabbit anti-activin receptor-like kinase (ALK3, 1:1000, YT5528), rabbit anti-Histone H3 (H3, 1:2000, YK0006), and rabbit anti-farnesoid X receptor (FXR, 1:1000, YN2161) were purchased from ImmunoWay Biotechnology (Suzhou, Jiangsu, China). Primary antibodies rabbit anti-activin receptor type 1 (ALK2, 1:1000, ab155981), rabbit anti-CREBH (1:1000, ab111938) and rabbit anti-pSTAT3 Ser727 (1:1000, ab32143) were purchased from Abcam (Cambridge, UK). Primary antibodies rabbit anti-TGR5 (1:1000, NBP2-23669) and rabbit anti-ferroportin (1:1000, NBP1-21502) were purchased from Novus Biologicals (Littleton, Colorado, USA). For protein quantity analysis in the whole cells, GAPDH was used as the housekeeping protein. For protein quantity analysis in cell nucleus, H3 was used as the housekeeping protein.

Horseradish peroxidase-labeled goat anti-rabbit IgG (H + L) (1:10,000, Beyotime Biotechnology, Beijing, China, A0208) and horseradish peroxidase-labeled goat anti-mouse IgG (H + L) (1:10,000, Beyotime Biotechnology, Beijing, China, A0216) were used as the secondary antibody respectively. The bands were quantified through Image Lab software (Amersham Imager 600, GE, Boston, MA, USA). 

### 2.7. Animal Treatment 

Forty male C57/BL6 mice (7–8 weeks of age, 20–25 g) were obtained from the Weitong Lihua Laboratory Animal Technology Company (Beijing, China) experimental animal center. The mice were housed under SPF conditions with free diet and water access. All mice were acclimated for a week before experimental treatment. Mice were equally assigned to four groups. The control group was administered 0.9% saline by gavage at a volume of 5 mL/kg/d. The GCDCA-exposed groups were administered 100 mg/kg/d, 200 mg/kg/d and 300 mg/kg/d GCDCA at the same volume, respectively. The treatment lasted 8 weeks, during which the body weight of mice was recorded weekly.

At the end of the experiments, mice were anesthetized with barbiturate (intraperitoneal injection, 150 mg/kg pentobarbital sodium). Then, 300 µL of blood was collected from the medial canthus of eye and centrifuged to obtain serum. In addition, another 20 μL of whole blood was collected in an anticoagulant tube for blood routine test. After that, the mice were euthanized and analyzed. All of the collected tissues and serum samples were stored at −80 °C for further testing. The experiments and protocols were approved by the Ethics Committee of China Agricultural University (Aw02402202-4-2).

### 2.8. Analysis of GCDCA Concentration in Mice

Serum samples (50 μL serum for each mouse) were shaken for 5–10 s and mixed with 10 μL of internal standard sample. Then, the mixture was added to 190 μL methanol solution and shaken for 30 s to extract the GCDCA in serum. Extracted GCDCA in methanol solution was gotten by centrifugation at 14,000× *g* for 10 min at 4 °C to separate the supernatant. Then, 200 μL of ultrapure water was added to the separated methanol solution containing GCDCA for freeze-drying. After that, the residue was re-dissolved in 200 μL ultrapure water. Each of the dissolved sample was centrifuged at 12,000× *g* for 5 min, and the supernatant was transferred into a liquid injection vial (including internal cannula) for HPLC analysis [5].

### 2.9. Analyses of Iron Parameters and Serum Hepcidin Level

Serum iron and liver non-heme iron in mice were measured according to the method as described previously [12]. Determination of serum hepcidin was performed using ELISA in 96-well plates coated with a polyclonal antibody against hepcidin (R&D Systems, Minneapolis, MN, USA). The assay procedures were performed according to the manufacturer’s instructions. 

### 2.10. Statistical Analyses

Results were expressed as means ± standard deviation (SD). Comparison between groups were analyzed using the Student’s *t*-test or one-way ANOVA with the LSD multiple range test, Tamhane’s T2 and Tukey’s test. Differences symbolled as *p* < 0.05, *p* < 0.01 and *p* < 0.001 were considered statistically significant at different levels. All procedures were conducted using the program SPSS 21.0 (SPSS, Inc., Chicago, IL, USA).

## 3. Results

### 3.1. Imbalance of Iron Homeostasis Occurred Simultaneously with Increased Serum GCDCA Level in Patients with CKD 

Patients with CKD are prone to imbalanced iron levels, which are manifested as abnormal blood routine indices. From our previous study in patients with CKD, it was found that the blood indicators including red blood cell (RBC), hemoglobin (HGB), hematocrit (HCT) and mean corpuscular hemoglobin concentration (MCHC) were all decreased significantly compared to those healthy subjects (*p* < 0.001, Table 1), accompanying with low level of serum iron. The level of serum iron was 12.89 ± 5.61 μmol/L (male) and 10.89 ± 4.39 μmol/L (female) in CKD patients in this study (Table 2), which was lower than those reported in healthy subjects (14.82 ± 0.72 μmol/L [13]). Further, the level of hepcidin in serum indicated that hepcidin in patients (43.85 ± 25.48 ng/mL) was increased by eight-fold compared to healthy individuals (5.29 ± 3.37 ng/mL, Appendix A), implying a similar trend as that observed in previous reports [9,10,11].

The concentrations of bile acids in these patients were also determined. From Table 3, it may be seen that the concentrations of conjugated bile acids in total detected bile acids increased significantly in patients (2290.00 ± 1720.81 μg/L for male and 1801.83 ± 1183.49 μg/L for female) compared to healthy subjects (1053.84 ± 550.55 μg/L for male and 840.59 ± 412.21 μg/L for female) (*p* < 0.001). Notably, the increase of GCDCA mostly contributed to the increase of conjugated bile acids. The concentration of GCDCA in healthy subjects was 690.48 ± 358.83 μg/L (male) and 501.15 ± 269.94 μg/L (female), while those in CKD patients increased to 1238.75 ± 1005.47 μg/L (male) and 983.74 ± 623.56 μg/L (female) (*p* < 0.001).

### 3.2. GCDCA Up-Regulated Hepcidin in a Dose-Related Manner

HepG2 cells were used to determine the effect of GCDCA in vitro. As shown in Figure 1A, the cell viabilities were above 85% of those in the control group when GCDCA concentrations were tested from 10 µM to 500 µM. When the concentrations of GCDCA increased further (750 µM and 1000 µM), the cell viabilities decreased to less than 70% of those of the control group.

Next, the dose and time effects of GCDCA on hepcidin were examined. It was revealed that GCDCA resulted in a dose-dependent increase in hepcidin expression, with the highest, i.e., by nearly 12-fold compared with control group, in response to 500 μM GCDCA. In addition, 200 µM and 300 µM GCDCA also up-regulated hepcidin significantly (*p* < 0.01, Figure 1B). Herein, the lowest GCDCA concentration (200 μM) in the following treatment was chosen. Then, the time effect of GCDCA on hepcidin expression was analyzed. The result indicated that hepcidin increased by more than two-fold compared to the control at 4-h GCDCA treatment time, which was higher than other treatments (Figure 1C).

Interleukin-6 (IL-6) is known as a potent inducer of hepcidin. Since a higher concentration of GCDCA triggered cytotoxicity (Figure 1A), we also determined the gene expression of IL-6 after GCDCA treatment. The results showed that the mRNA level of IL-6 in GCDCA-treated cells was similar with control values (*p* > 0.05), indicating that GCDCA treatment did not affect the expression of IL-6 (Figure 1D). Thus, it could be speculated that GCDCA induced hepcidin expression directly in vitro.

### 3.3. BMP6/ALK3-SMAD Signaling Mainly Mediated GCDCA-Induced Hepcidin Expression

To clarify how GCDCA activated the transcription of hepcidin, we examined the total expression and nuclear expression of three related markers in the upstream pathways for hepcidin regulation, i.e., SMAD1/5/8, STAT3 and CREBH. GCDCA significantly enhanced the phosphorylation of SMAD1/5/8 at 2 h (*p* < 0.01), while no remarkable effects were observed on STAT3 phosphorylation throughout the four hours of treatment (*p* > 0.05, Figure 2A,B). Elevated phosphorylation of SMAD1/5/8 in nuclear protein was further detected. Similarly, another transcriptional factor of hepcidin, CREBH, was not changed in nuclear (*p* > 0.05, Figure 2C). In addition, inhibiting the effect on the phosphorylation of SMAD1/5/8 by LDN-193189 relieved GCDCA-induced hepcidin gene expression (Figure 2D) and phosphorylation of SMAD1/5/8 (Figure 2E), which indicated that SMAD signaling plays an important role in GCDCA-induced hepcidin regulation in vitro.

Regulatory factors upstream of SMAD1/5/8, including BMPs (BMP2, BMP4 and BMP6), BMP receptors (ALK2, ALK3, BMPRIIA and BMPRIIB), and BMP coreceptor hemojuvelin (HJV), were tested. Among these factors, GCDCA significantly up-regulated the expression of BMP6 and ALK3 (*p* < 0.05), in terms of both gene and protein expression levels (Figure 3). The protein level of BMP2 decreased after GCDCA treatment. However, no significant changes were observed among other BMP subtypes, BMP receptors or the BMP coreceptor. These results suggest that GCDCA stimulates hepcidin expression through the BMP6/ALK3-SMAD pathway in vitro.

### 3.4. FXR Was Triggered by GCDCA for Hepcidin Expression

The underlying mechanisms of GCDCA-induced BMP6 up-regulation and hepcidin expression were further investigated. GCDCA could be identified by receptors such as FXR and TGR5, which will activate the several metabolic signaling responses in cells and tissues. Our in vitro data suggest that GCDCA treatment significantly activates FXR expression (by about 1.5 fold compared to the control at the mRNA level and by 1.8 fold at the protein level) (*p* < 0.05). The expression of TGR5 seemed to be not obviously changed in the GCDCA-treated groups compared to the control group (*p* > 0.05, Figure 4A,B).

To further validate the role of FXR in GCDCA-induced hepcidin up-regulation, GW4064 (an activator of FXR) was used to treat HepG2 cells. The results showed that GW4064 promoted the over-expression of FXR, leading to an increase of Hamp expression and phosphorylation of SMAD1/5/8 (Figure 4C,D). Additionally, an inhibitive intervention of FXR using guggulsterone on GCDCA-treated HepG2 cells showed that guggulsterone significantly prevented GCDCA-induced FXR activation (*p* < 0.001) and elevation of hepcidin (*p* < 0.001) (Figure 4E,H). The expression of BMP6 and ALK3 induced by GCDCA was also inhibited (Figure 4F,G). Thus, GCDCA induced hepcidin expression via FXR-BMP6/ALK3 mediated signaling. These results further confirm the role of FXR activation in triggering GCDCA-induced hepcidin expression through SMAD signaling.

### 3.5. GCDCA Disturbed Iron Homeostasis in Mice

In order to investigate whether GCDCA could disrupt iron homeostasis in vivo, the effect of GCDCA on fundamental indicators (Appendix A) and iron related parameters in C57/BL6 mice was investigated. Firstly, the serum concentration of GCDCA was measured. In the control group, the concentration of GCDCA was 2.90 ± 0.87 µg/L. In 100 mg/kg/d, 200 mg/kg/d, and 300 mg/kg/d GCDCA-treated groups, the GCDCA concentration in serum was 8.16 ± 2.49 µg/L, 14.95 ± 1.7 µg/L, and 23.84 ± 7.63 µg/L, respectively (Figure 5A). This result indicated that GCDCA concentration in serum increased response to increased concentrations of GCDCA treatments. Phenotypically, this demonstrated a potent effect of GCDCA in terms of reducing blood routine indices such as HGB, HCT, RBC, MCHC and mean corpuscular hemoglobin (MCH) (Figure 5B–F). These results indicate that mice treated with high doses of GCDCA tend to be anemic, consistent with what is observed in patients with CKD.

The concentration of serum iron decreased about 30% in mice treated with 300 mg/kg/d GCDCA compared to control (*p* < 0.001, Figure 5G). The mRNA level of divalent metal transporter 1 (DMT1) in duodenum increased significantly compared to control (*p* < 0.05, Figure 5H). Additionally, the protein level of ferroportin in duodenum displayed a decreased tendency after GCDCA treatment at 300 mg/kg/d (*p* < 0.01, Figure 5I), indicating an inhibition of iron release from duodenum to blood. Additionally, the level of iron stored in liver was increased after 200 mg/kg/d and 300 mg/kg/d GCDCA treatment (Figure 5J). The protein expression of ferroportin in liver was significantly down-regulated in 200 mg/kg/d and 300 mg/kg/d GCDCA treatment groups compared to control (*p* < 0.05, Figure 5K), indicating an inhibition of iron release from liver.

### 3.6. GCDCA Activated FXR-SMAD Signaling for Hepcidin Up-Regulation in Mice

Similarly, serum hepcidin level and hepatic Hamp gene expression were up-regulated in 200 mg/kg/d and 300 mg/kg/d GCDCA-treated mice compared to the control (*p* < 0.01, Figure 6A,B). The level of inflammatory factors in liver was measured, including the mRNA levels of IL-6 and tumor necrosis factor-α (TNF-α). None of the indicators displayed any differences among all of the experimental groups, implying that GCDCA treatment did not cause inflammatory response in mice (*p* > 0.05, Figure 6C,D). Similar to the result we observed in vitro, the phosphorylation of SMAD1/5/8 also increased in liver of GCDCA-treated mice (*p* < 0.05, Figure 6E), with no significant changes for phosphorylated STAT3 expression relative to control (*p* > 0.05, Figure 6E). The protein level of FXR was also elevated under 200 mg/kg/d and 300 mg/kg/d GCDCA treatment groups (*p* < 0.05, Figure 6E). These findings verified the FXR-SMAD signaling pathway mediated hepcidin up-regulation under GCDCA treatment in vivo. Thus, high level of GCDCA could induce excessive hepcidin expression, which was mainly mediated by FXR-SMAD signaling pathway.

## 4. Discussion

Hepcidin, a liver-derived peptide hormone, is the key regulator of systemic iron homeostasis. Dysregulated hepcidin is involved in the pathophysiology of many iron-related diseases, including anemia in patients with CKD [14,15]. Notably, except for inflammation and erythropoietin deficiency, there are some other important regulatory factors for hepcidin expression. To clearly state the hepcidin regulatory pathway is vital for finding strategies with which to treat patients with such iron-related disorders. From our population experiment, it was found that the serum iron level decreased in patients, while hepcidin concentration increased significantly (by more than eight-fold) in patients with CKD. Interestingly, the serum composition of bile acids was also disturbed in these patients. Among them, the ratio of conjugated bile acids was up-regulated, with GCDCA contributing the most. GCDCA is a kind of bile acid which belongs to the steroid family. In patients with CKD, the serum concentration of GCDCA was found to be increased compared to healthy controls. Previous studies have shown that steroids are related to iron homeostasis [16]. A lot of steroids in mammals are known to affect iron levels through regulating serum concentrations of hepcidin. An increase of serum cholesterol content was positively correlated with hepcidin level, causing increased serum ferritin and iron disorder in kidney transplant patients [17,18]. The increase of serum ferritin is an indicator of excessive iron storage in body (i.e., iron stored in liver) induced by hepcidin. Similarly, the phenomenon of elevated hepcidin was also found simultaneously with increased serum ferritin and disordered iron homeostasis in our study. Further, hepcidin expression was abnormal in cholestatic cirrhosis patients, indicating a possible regulatory effect of hepcidin due to elevated bile acid levels [7]. Elevated hepcidin levels in patients with sickle cell disease could exacerbate anemia, and patients treated with steroids showed an increase in serum hepcidin and a decrease in serum erythronferrone (a glycoprotein hormone which could inhibit hepcidin synthesis) [19]. Estrogen was also found to be able to up-regulate hepcidin and decrease iron absorption in ovariectomized mice [20]. Thus, we speculated that GCDCA might have regulatory effect on hepcidin.

The experiments in vitro performed on HepG2 cells was used to elucidate the effect of GCDCA on hepcidin. GCDCA dose-dependently increased hepcidin gene expression in vitro. In addition, hepcidin increased by more than two-fold compared to the control when treated with GCDCA for 4 h. Additionally, IL-6 was a potent inducer of hepcidin [2]. In order to figure out whether GCDCA directly caused the increase of hepcidin expression or via regulating IL-6 related pathway, we also determined IL-6 expression in HepG2 cells under GCDCA treatment. It was found that treatment of GCDCA did not affect IL-6 expression significantly. Thus, it indicated that GCDCA increased hepcidin expression directly. Additionally, the hepcidin up-regulation effect was also found on another steroid, estrogen, using HepG2 cell model [20]. Overall, it implied that GCDCA possessed effect on elevating hepcidin expression.

The regulation of hepcidin is mainly performed on a transcription level. Previous studies suggested that SMAD1/5/8 phosphorylation and STAT3 phosphorylation were related signaling pathways which could induce hepcidin transcriptional expression [21,22]. The expression of STAT3 is related to inflammatory response, and the activation of the STAT3 pathway regulated by inflammatory factors could stimulate the expression of hepcidin. In addition, researchers also proved that endoplasmic reticulum stress regulated hepcidin by one of its products CREBH [23]. It is a regulator for hepcidin expression which acts via binding at the hepcidin promoter [24]. In this study, it was found that GCDCA treatment up-regulated the phosphorylation of SMAD1/5/8. SMAD1 was the main subtype of SMADs in hepatocytes, and usually taken as the housekeeping protein to investigate the phosphorylated condition of SMAD1/5/8 (an important regulatory signaling of hepcidin) [21]. However, no remarkable changes occurred on STAT3 phosphorylation nor CREBH expression in HepG2 cells. Thus, it was further verified that either STAT3 or CREBH related signaling was the main functional pathway for GCDCA-induced hepcidin expression. Moreover, the inhibitor of SMAD1/5/8 phosphorylation LDN-193189 completely blocked the phosphorylation signaling of SMAD1/5/8 and prevented hepcidin induction by GCDCA. These findings demonstrate the essential role of the SMAD signaling pathway in GCDCA-regulated hepcidin induction.

Recent studies have shown that BMPs activate the regulatory pathway for hepcidin expression [25,26,27]. After BMPs bind to the receptor on cell membrane, it transmits the phosphorylation signal to regulate SMAD factor, and then activates the transcription of hepcidin in the nucleus [25]. The BMP receptors (termed as activin receptor-like kinases, ALK) responsible for hepcidin up-regulation are type I (BMPRI, also named ALK2 and ALK3) and type II (BMPRII) receptors, respectively [21]. In this study, it was found that GCDCA treatment significantly up-regulated the protein level of BMP6, as well as weakly down-regulation of BMP2. Additionally, the expression of ALK3 was found increased after GCDCA treatment. The BMP/SMAD pathway has two branches. One is the regulating of hepcidin activation by ALK3 in response to BMP2 or BMP6. The other one is increased hepcidin expression by ALK2-dependent activation of BMP6 [21]. Since ALK3 could be activated by BMP2 or BMP6 ligands, BMP2 displayed a downward trend in response to the signal changes after activating BMP6 with GCDCA. Taken together, it was demonstrated that GCDCA up-regulated hepcidin expression mainly through activating the BMP6/ALK3-SMAD signaling pathway.

Farnesoid X receptor (FXR, NR1H4) is a nuclear hormone receptor activated by endogenous bile acids, and its role in regulating liver metabolic homeostasis has been extensively studied. In addition, FXR is the major identifying nuclear receptor for GCDCA [28,29]. In this study, GCDCA treatment activated FXR and brought excessive synthesis of hepcidin. In iron overloaded mouse model, FXR activation was found to regulate hepcidin expression, so as to maintain iron homeostasis and protect the body from iron toxicity [30]. Thus, it could be further inferred that the activation of FXR could up-regulate hepcidin expression to decrease body iron level. In addition, some of the regulatory actions of bile acids are also mediated through activating cell surface G protein-coupled receptors, including the transmembrane G protein-coupled receptor TGR5 [31]. Thus, the protein level of TGR5 was also determined. However, there was no remarkable changes of TGR5 in our study. Therefore, FXR became a vital target for GCDCA-induced hepcidin expression. In order to further validate it, we also performed an FXR activator experiment using GW4064. It is a known FXR agonist which has similar nuclear parent with GCDCA [32]. Our result indicated that GW4064 also up-regulated hepcidin expression via the same major pathway of FXR-SMAD-related signaling. Additionally, intervention treatment with FXR inhibitor (guggulsterone) significantly prevented GCDCA-induced FXR activation and hepcidin expression. These findings brought up a potential new avenue that increased GCDCA was responsible for hepcidin regulation via FXR mediated SMAD signaling pathway.

Finally, the effect of GCDCA on FXR-SMAD signaling-related hepcidin expression and disturbed iron homeostasis were verified in C57/BL6 mice. Our results demonstrated a significant effect of GCDCA on reducing serum iron in mice. One of the major functions of iron is to participate in synthesizing HGB in red blood cell [33]. Consistently, GCDCA treatment decreased the serum iron and led to remarkably down-regulated HGB. When iron is deficient in the body, iron absorption increased manifesting as significantly increased intestinal DMT1 expression. In this study, the DMT1 expression in duodenum was increased after GCDCA treatment. Additionally, the protein level of duodenum ferroportin was found decreased after GCDCA treatment. Ferroportin was responsible for iron release from tissues (duodenum, liver, and macrophages), which was controlled by hepcidin [34]. The liver is the major organ to produce and secrete hepcidin. In this study, elevated hepcidin induced by GCDCA also degraded ferroportin, leading to the storage of excessive amounts of iron in liver. These results verified the effects of GCDCA on up-regulating hepcidin expression, resulting in a lack of iron homeostasis and even anemic tendency. To verify the signaling that GCDCA acted on hepcidin up-regulation, phosphorylation of SMAD and STAT3 in liver were tested. Results indicated an increased phosphorylation of SMAD with no remarkable changes on STAT3, a similar phenomenon to what we have gotten in vitro. Consistently, it was also un-conspicuous for the changes of inflammatory factors IL-6 and TNF-α expression (upward regulators for STAT3) in GCDCA treated mice compared to control. In addition, the protein level of FXR in liver was also increased after GCDCA treatment. Thus, it was further verified in vivo that FXR-SMAD related signaling was the main functional pathway for GCDCA-induced hepcidin expression.

Since the expression of FXR is occurred in many organs including liver and kidney, it is essential to maintain biological homeostasis in the whole body. Results of this study indicated that FXR activation promoted hepcidin expression and led to disturbed iron homeostasis. In some reports, targeting on FXR activation could be used to prevent the progression of inflammation and fibrosis in liver and kidney disease. Thus, the activation of FXR might have a dual effect. Interestingly, the subtypes of FXR distinctively existed in liver and kidney, and the main mechanism of iron homeostasis imbalance in nephropathy patients is related to hepatic FXR activation. Therefore, we concluded that targeting the expression of FXR subtypes in different organs may achieve the regulation for different disease. Further research on the mechanism of hepatic FXR subtype activation in GCDCA-induced hepcidin expression would be helpful to better interpret the regulation of iron homeostasis in patients with nephropathy.

## 5. Conclusions

An imbalance of iron homeostasis occurred in patients with CKD, accompanied with up-regulated hepcidin in serum. In addition, increased concentration of GCDCA was found. Excessive GCDCA could lead to increased expression of hepcidin via the BMP6/ALK3-SMAD signaling pathway in HepG2 cells. FXR was the receptor that GCDCA mainly targeted for hepcidin regulation. This FXR-SMAD signaling related hepcidin elevation was verified in mice under GCDCA treatment. The imbalance of GCDCA resulted in the deficiency of serum iron in vivo, which gives rise to potential risk for iron related diseases.

## Figures and Tables

**Figure 1 nutrients-14-03176-f001:**
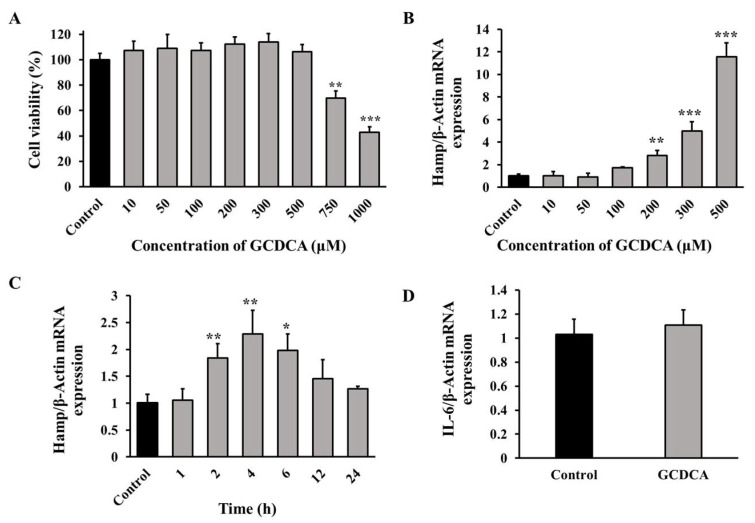
Effects of GCDCA on cell viability and hepcidin expression in HepG2 cells. (**A**) Cell viabilities treated with GCDCA at different concentrations. (**B**) Effect of GCDCA on hepcidin expression at concentrations of 10, 50, 100, 200, 300 and 500 µM for 24 h. (**C**) Hepcidin expression after 200 µM of GCDCA treatment for different treating time. (**D**) The effect of GCDCA on interleukin-6 (IL-6) expression treating at 200 µM for 4 h. The results were displayed as relative expression compared to control group. Values were shown as means ± SD. *, *p* < 0.05, GCDCA treatment group vs. control group. **, *p* < 0.01. ***, *p* < 0.001. Each experiment was performed at least three times.

**Figure 2 nutrients-14-03176-f002:**
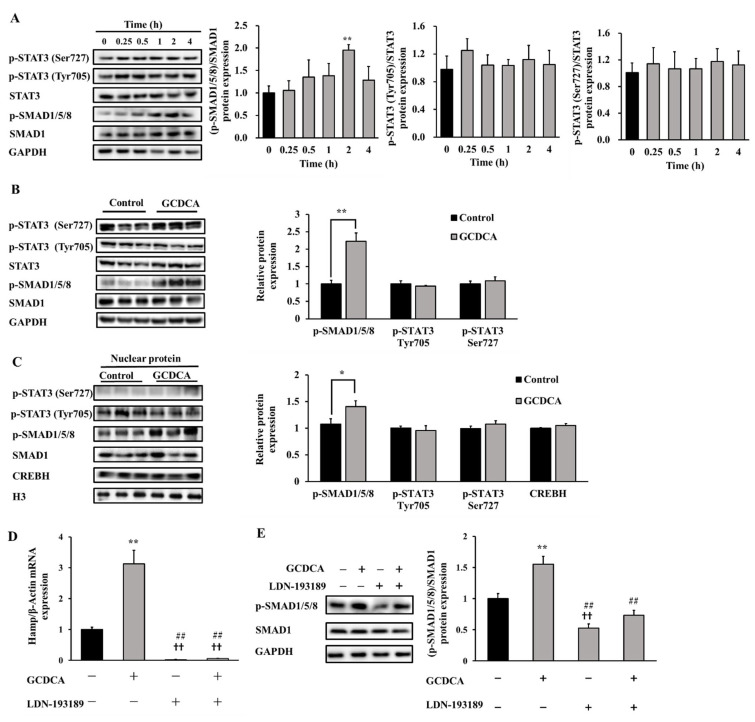
Changes of signaling proteins responsible for hepcidin expression after GCDCA treatment in HepG2 cells. (**A**) Representative western blotting bands shown in the left side of panel of A display the phosphorylation of SMAD1/5/8 (normalized by total SMAD1) and STAT3 (normalized by total STAT3) in response to different GCDCA treating time. Fold changes in optical density relative to controls are shown on the right side of panel of (**A**). (**B**) The phosphorylated and total protein levels of both SMAD1/5/8 and STAT3 after 2 h treatment with GCDCA. (**C**) The phosphorylated and total levels of SMAD1/5/8, STAT3 and CREBH in the nucleus after a 2-h treatment with GCDCA. (**D**) The hepcidin gene expression under co-treatment of 200 µM GCDCA and 150 nM LDN-193189 (SMAD inhibitor) for 2 h. (**E**) Representative western blotting bands showed the phosphorylation levels of SMAD1/5/8 under co-treatment of GCDCA and LDN-193189. Values were shown as means ± SD. *, *p* < 0.05, GCDCA treatment group vs. control group. **, *p* < 0.01. ††, *p* < 0.01, other experimental groups vs. control group. ##, *p* < 0.01, GCDCA treatment group vs. other experimental groups. Each experiment was performed at least three times.

**Figure 3 nutrients-14-03176-f003:**
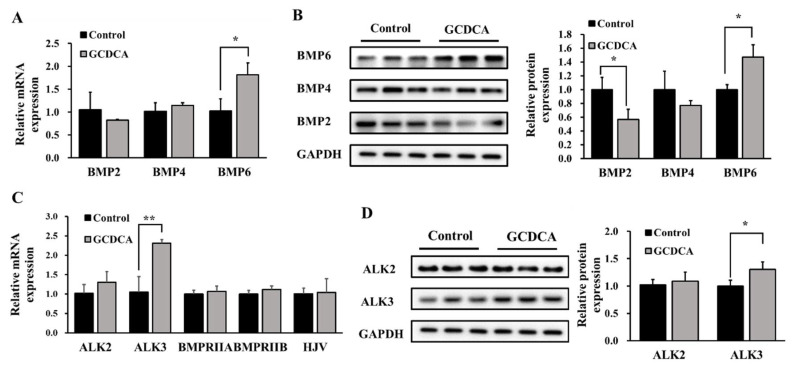
Effects of GCDCA on the changes of signaling proteins that responsible for the phosphorylation of SMAD1/5/8 in HepG2 cells. (**A**) Relative gene expression of BMP2, BMP4 and BMP6 after GCDCA treatment. (**B**) Representative western blotting bands of the protein levels of BMP2, BMP4 and BMP6 under GCDCA treatment, as well as their fold changes in optical density relative to controls. (**C**) Relative gene expression of ALK2, ALK3, BMPRIIA, BMPRIIB and HJV after GCDCA treatment. (**D**) Effects of GCDCA on protein expressions of ALK2 and ALK3, as well as their fold changes in optical density relative to controls. Values were shown as means ± SD. *, *p* < 0.05, GCDCA treatment group vs. control group. **, *p* < 0.01. Each experiment was performed at least three times.

**Figure 4 nutrients-14-03176-f004:**
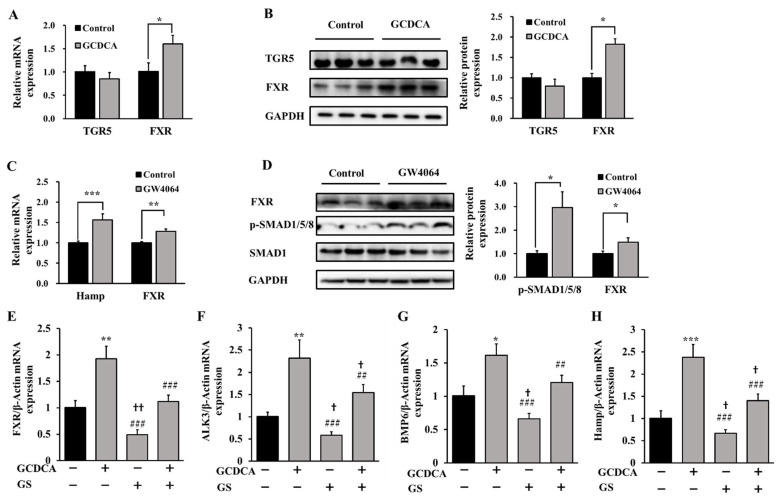
Effects of GCDCA on the changes of BMP6/ALK3 signaling related receptors in HepG2 cells. (**A**) Relative gene expression of TGR5 and FXR after GCDCA treatment. (**B**) Representative western blotting bands of TGR5 and FXR under GCDCA treatment, as well as their fold changes in optical density relative to controls. (**C**) Relative gene expression of hepcidin and FXR after treatment of a FXR activator (GW4064, 10 µM for 2 h). (**D**) The effects of GW4064 on the phosphorylation of SMAD1/5/8 (normalized by total SMAD1) and protein level of FXR. (**E**–**H**) The effect of guggulsterone (GS, a FXR inhibitor, pretreated HepG2 cells for 18 h at 20 µM) on GCDCA-induced activation of FXR-BMP6/ALK3 signaling proteins and hepcidin expression. Values were shown as means ± SD. *, *p* < 0.05, GCDCA treatment group vs. control group. **, *p* < 0.01. ***, *p* < 0.001. †, *p* < 0.05, other experimental groups vs. control group. ††, *p* < 0.01. ##, *p* < 0.01, GCDCA treatment group vs. other experimental groups. ###, *p* < 0.001. Each experiment was performed at least three times.

**Figure 5 nutrients-14-03176-f005:**
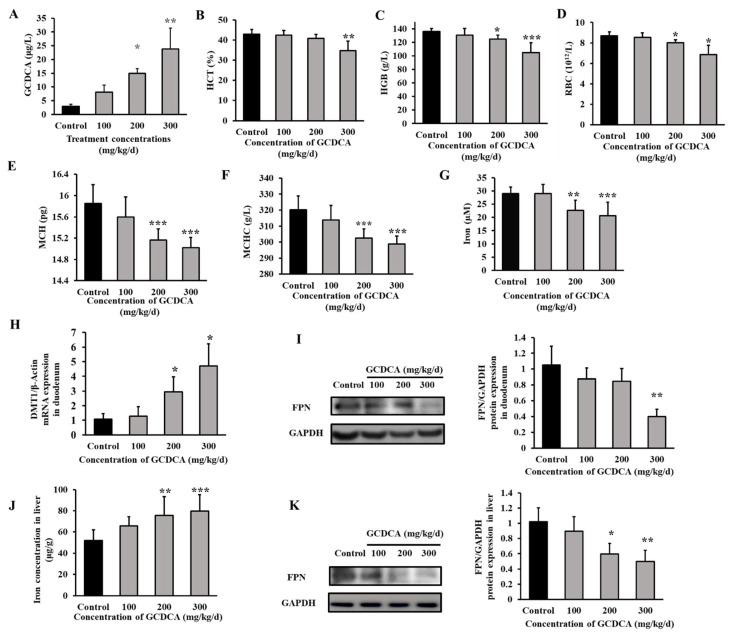
The serum concentrations of GCDCA, changes of blood routine indices, and iron related indicators in C57BL/6 mice treated with GCDCA at concentrations of 0, 100 mg/kg/d, 200 mg/kg/d and 300 mg/kg/d for 8 weeks (*n* = 10). (**A**) GCDCA concentrations in serum (**B**) Hematocrit, HCT. (**C**) Hemoglobin in blood, HGB. (**D**) Red blood cell, RBC. (**E**) Mean corpuscular hemoglobin, MCH. (**F**) Mean corpuscular hemoglobin concentration, MCHC. (**G**) Serum iron level. (**H**) Gene expression of DMT1 in duodenum. (**I**) The protein expression of ferroportin (FPN) in duodenum. (**J**) Iron level in liver. (**K**) The protein expression of FPN in liver. Values were shown as means ± SD. *, *p* < 0.05, GCDCA treatment group vs. control group. **, *p* < 0.01. ***, *p* < 0.001. Each experiment was performed at least three times.

**Figure 6 nutrients-14-03176-f006:**
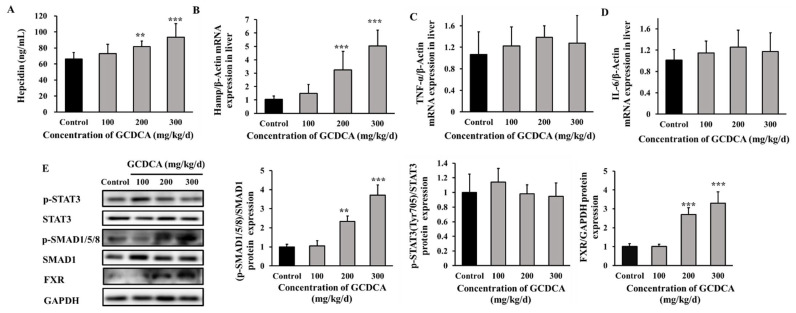
The possible action mechanism underlying GCDCA-induced iron decrease in mice. (**A**) Concentration of serum hepcidin. (**B**) Gene expression of hepcidin in liver. (**C**,**D**) Gene expression of inflammatory factors (TNF-α and IL-6) in liver. (**E**) The levels of phosphorylated SMAD1/5/8 and STAT3, as well as the FXR protein level in liver. Values were shown as means ± SD. **, *p* < 0.01, GCDCA treatment group vs. control group. ***, *p* < 0.001. Each experiment was performed at least three times.

**Table 1 nutrients-14-03176-t001:** Red blood cell indices of 223 patients with CKD and 69 healthy subjects.

	Male			Female		
Indicators	Healthy (*n* = 37)	CKD (*n* = 147)	*p*-Value	Healthy (*n* = 32)	CKD (*n* = 76)	*p*-Value
RBC (10^12^/L)	4.98 ± 0.43	3.55 ± 0.49 ***	<0.001	4.53 ± 0.21	3.48 ± 0.4 ***	<0.001
HGB (g/L)	151.00 ± 11.03	108.64 ± 15.81 ***	<0.001	133.91 ± 7.92	105.03 ± 12.97 ***	<0.001
HCT (%)	45.90 ± 3.74	33.85 ± 4.52 ***	<0.001	40.87 ± 1.95	32.88 ± 4.04 ***	<0.001
MCH (g/L)	30.93 ± 1.31	30.21 ± 2.12 *	0.0137	29.59 ± 1.49	30.19 ± 1.49	0.0701
MCHC (pg)	336.05 ± 9.33	325.98 ± 13.32 ***	<0.001	327.56 ± 6.4	319.92 ± 11.37 ***	<0.001
MCV (fL)	92.14 ± 5.04	93 ± 6.28	0.3810	90.3 ± 3.48	94.71 ± 5.01 ***	<0.001

CKD, chronic kidney disease, RBC, red blood cell; HGB, hemoglobin in blood; HCT, hematocrit; MCH, mean corpuscular hemoglobin; MCHC, mean corpuscular hemoglobin concentration; MCV, mean corpuscular volume. The symbol * indicates a significance of *p* < 0.05 between CKD patients and healthy subjects, and *** indicates a significance of *p* < 0.001.

**Table 2 nutrients-14-03176-t002:** The iron related parameters in patients with CKD.

Indicators	Fe (μmol/L)	FERR (ng/mL)	TIBC (μmol/L)	UIBC (μmol/L)	TSAT (%)	TRF (g/L)
Male	12.89 ± 5.61	349.91 ± 418.82	45.75 ± 10.78	31.38 ± 10.49	29.5 ± 14.14	1.89 ± 0.52
Female	10.89 ± 4.39	403.44 ± 291.04	44.55 ± 12.3	29.81 ± 9.29	26.25 ± 11.31	1.84 ± 0.51

Fe, Serum iron; FERR, ferritin; TIBC, total iron binding capacity; UIBC, unsaturated iron-bonding capacity; TSAT, Transferrin saturation; TRF, transferrin.

**Table 3 nutrients-14-03176-t003:** Concentrations of bile acids (µg/L) in CKD patients and healthy subjects.

	Male			Female		
Bile Acids	Healthy (*n* = 37) (µg/L)	CKD (*n* = 147) (µg/L)	*p*-Value	Healthy (*n* = 32) (µg/L)	CKD (*n* = 76) (µg/L)	*p*-Value
Glycochenodeoxycholate	690.48 ± 358.83	1238.75 ± 1005.47 ***	<0.001	501.15 ± 269.94	983.74 ± 623.56 ***	<0.001
Chenodeoxycholic acid	508.63 ± 355.17	186.53 ± 185.29 ***	<0.001	298.42 ± 125.93	143.43 ± 126.36 ***	<0.001
Deoxycholate	437.25 ± 294.28	129.36 ± 103.56 ***	<0.001	233.16 ± 91.1	125.92 ± 117.79 ***	<0.001
Glycodeoxycholate	160.23 ± 118.07	364.68 ± 387.49 ***	<0.001	77.97 ± 62.73	214.48 ± 236.24 ***	<0.001
Glycoursodeoxycholic acid	164.60 ± 112.96	189.18 ± 247.74	0.3949	100.66 ± 133.89	241.88 ± 308.93 **	0.0018
Cholic acid	108.62 ± 139.56	41.57 ± 61.98 *	0.020	45.10 ± 28.06	24.63 ± 17.86 **	0.0022
Glycocholic acid	104.68 ± 86.29	239.67 ± 263.58 ***	<0.001	77.97 ± 62.73	214.48 ± 236.24 ***	<0.001
Ursodeoxycholic acid	70.96 ± 55.96	34.25 ± 39.02 **	0.0028	34.31 ± 23.71	36.13 ± 46.22	0.8037
Taurochenodeoxycholate	52.38 ± 66.25	224.17 ± 297.08 ***	<0.001	46.55 ± 24.59	129.01 ± 118.69 ***	<0.001
Glycochenodeoxycholate 3-glucuronide	39.46 ± 22.75	60.77 ± 64.43 **	0.0017	22.41 ± 15.31	37.25 ± 48.82 *	0.0263
Taurodeoxycholate	14.86 ± 21.46	87.21 ± 140.56 ***	<0.001	29.70 ± 20.77	68.40 ± 82.71 ***	<0.001
Taurocholic acid	9.58 ± 9.97	67.15 ± 92.9 ***	<0.001	10.98 ± 5.27	39.34 ± 46.04 ***	<0.001
Tauroursodeoxycholate	6.38 ± 5.38	15.33 ± 18.95 ***	<0.001	2.88 ± 1.73	14.63 ± 16.29 ***	<0.001
Conjugated bile acids	1053.84 ± 550.55	2290.00 ± 1720.81 ***	<0.001	840.59 ± 412.21	1801.83 ± 1183.49 ***	<0.001
Unconjugated bile acids	794.52 ± 316.46	364.97 ± 257.36 ***	<0.001	553.15 ± 219.42	289.46 ± 169.32 ***	<0.001
Conjugated bile acids/Unconjugated bile acids	1.35 ± 0.83	7.99 ± 6.57 ***	<0.001	1.85 ± 1.4	7.34 ± 5.76 ***	<0.001

The symbol * indicates a significance of *p* < 0.05 between CKD patients and healthy controls. ** indicates a significance of *p* < 0.01, and *** indicates significance of *p* < 0.001.

## Data Availability

Data presented in this study are available on request from the corresponding author.

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
