# Peer review of "Glycochenodeoxycholate Affects Iron Homeostasis via Up-Regulating Hepcidin Expression"

_nutrients, 2022, doi:10.3390/nu14153176_

Round 1

Reviewer 1 Report

Wang, et al. have showed that GCDCA, a member of steroids, was increased in patients with CKD compared to healthy subjects. In basic experiments, GCDCA augmented hepcidin expression through FXR-BMP6/ALK3-Smad-dependent pathway in HepG2 cells. Similarly, mice with GCDCA treatment also showed increase of hepcidin via FXR-Smad pathway with anemia. I concern that there are several critical issues in the present data.

Comments;

1)   In table 3, the values of bile acids were composition ratio but not concentration. The authors should show concentration of them in both healthy and CKD.

2)   The concentration of GCDCA correlated to hepcidin concentration in healthy and CKD subjects ?

3)   How did the authors decide the concentration of GCDCA in experiments ? The concentration of 200µM consist with that of CKD patients ? There is no data about it.

4)   GCDCA actually increased FXR expression, however, there was no evidence of GCDCA on hepcidin expression via FXR-dependent manner. The auttors should examine whether FXR siRNA or FXR inhibitor cancel the effect of GCDCA on BMP6/ALK3-hepcidin expression.

5)   In mice experiments, the concentration of CGDCA should be measured and demonstrated.

6)   Hepcidin normally internalizes and diminishes ferroportin expression, leading to the suppression of iron availability and absorption. Therefore, the authors should examine FPN protein expression in both spleen (or liver) and duodenum.

Reviewer 2 Report

The nutrients-1819816 addresses the alteration of bile acid in CKD and tries to explain the possible hepatic signals responded to glycochenodeoxycholate. The hepatic hepcidin regulation by glycochenodeoxycholate may have the novelty of this study; however, authors should provide additional and corrective information carefully.

1. [L127] at least three times – please specify the power in the individual figures.

2. [Table 1] Please match the effective digits.

3. [Table 2] If the data were adopted, authors had better cite them properly in the text but not in a table format.

4. [Table 2] If authors generated these data, please discrete male and female data as presented in Table 1 since sex is an important factor for iron homeostasis.

5. [Table 3] Please divide as male/female data.

6. [Table 3] Please provide standard deviations. Without standard deviations, statistical analysis may not apply.

7. [Table 3] Please see the spells of Bas (i.e. Glycochenodeoxycholate-3-glucu).

8. [Table 3] Con BAs have P-value of 0.000, but there is no star marked.

9. [Table 3] No stat for Con Bas/Uncon Bas.

10. [L192] 12-folds – 12-fold

11. [Figure 2] As seen in Figure 1, a higher concentration of GCDCA triggered cytotoxicity, and IL-6 is a potent inducer of hepcidin; however, the authors did not provide IL-6-related signals in Figure 2. Without IL-6, the induction of hepcidin by GCDCA may not be fully explainable.

12. [Figure 2] Please provide housekeeping signals for individual blots in the y-axis and/or materials and methods.

13. [Figure 2] Does SMAD1 is a suitable reference for p-SMAD1/5/8?

14. [L253] it is hard to say “regulate SMAD phosphorylation”.

15. [Figure 6] What are the housekeeping proteins?

16. Professional English correction is required.

17. Too many typos and grammatical errors are found.

18. Please provide a better resolution of the figures.
